# Late Diagnosis of Disseminated *Sporothrix brasiliensis* Infection with Bone Marrow Involvement in an HIV-Negative Patient

**DOI:** 10.3390/pathogens11121516

**Published:** 2022-12-10

**Authors:** Vanessa Caroline Randi Magalhães, Salene Angelini Colombo, Gustavo José Cota Freitas, Alexandre Sampaio Moura, Flávia Cardoso Lopez Vieira, Ana Cláudia Lyon, Maria Isabel Azevedo, Nalu Teixeira de Aguiar Peres, Daniel Assis Santos

**Affiliations:** 1Department of Microbiology, Institute of Biological Sciences, Universidade Federal de Minas Gerais, Av. Antônio Carlos, 6627, Belo Horizonte 31270-901, Brazil; 2Hospital Eduardo de Menezes, Fundação Hospitalar do Estado de Minas Gerais (FHEMIG), R. Dr. Cristiano Rezende, 2213, Belo Horizonte 30622-020, Brazil; 3Department of Preventive Veterinary Medicine, School of Veterinary, Universidade Federal de Minas Gerais, Av. Antônio Carlos, 6627, Belo Horizonte 31270-901, Brazil

**Keywords:** systemic sporotrichosis, pyoderma gangrenosum, hemophagocytic syndrome

## Abstract

Sporotrichosis is a fungal disease that causes symptoms similar to those of other infectious and non-infectious diseases, making diagnosis difficult and challenging. Here, we report a case of an HIV-negative patient presenting disseminated sporotrichosis with widespread cutaneous lesions mimicking pyoderma gangrenosum, with bone marrow infection, pancytopenia, and hemophagocytic syndrome. However, all the clinical manifestations and a bacterial coinfection delayed the request for a fungal diagnosis. Therefore, sporotrichosis should always be investigated in patients from endemic areas presenting with widespread cutaneous lesions associated with pancytopenia.

## 1. Introduction

*Sporothrix brasiliensis* is the leading agent of human and animal sporotrichosis in Brazil. The disease occurs after fungal inoculation on the host’s skin after being bitten or scratched by infected animals, mainly domestic cats [1]. On the other hand, *Sporothrix* species may also reach human patients through the traumatic implantation of the skin with fungal conidia from soil, plants, and decaying organic matter [2,3,4]. Sporotrichosis is also caused by other species, such as *Sporothrix schenckii*, *Sporothrix globosa*, and *Sporothrix luriei* [4,5]. However, *S. brasiliensis* is the most prevalent in Brazil, particularly in the south, southeast, and northeast [4,5].

The disease usually manifests as a chronic skin and subcutaneous infection with local lymph node involvement [3]. However, severe systemic organ manifestations have been observed in Brazil, mainly in immunocompromised individuals [6,7]. Low levels of CD4^+^ T cells in HIV patients predispose these individuals to the systemic dissemination of sporotrichosis. Other conditions such as diabetes, alcoholism, granulomatous diseases, cirrhosis, kidney transplantation, and chronic use of corticosteroids are also described to be associated with sporotrichosis [2,6].

In a recent systematic review of cases of sporotrichosis in Brazil carried out by Rabello et al. (2022) [5], the lymphocutaneous clinical form was predominant in 56.1% of the cases, followed by 27.1% of fixed cutaneous and 14.3% of systemic sporotrichosis. In that study, 16.9% of the patients were co-infected with HIV, 13.4% were diabetic, 6.1% were alcoholics, and 2.3% were smokers [5]. Despite its increasing incidence, severe sporotrichosis remains a diagnostic challenge, mainly due to its resemblance to other diseases. Here, we report a case of systemic sporotrichosis late diagnosis, favoring bone marrow infection in an HIV-negative patient. This study was approved by the Ethics Committee of Universidade Federal de Minas Gerais (CAAE 00883118.0.0000.5149) and Hospital Eduardo de Menezes—FHEMIG (CAAE 08003118.03001.5124).

## 2. Case Report

A 49-year-old female patient was admitted to an infectious diseases reference hospital in January 2020 in Belo Horizonte, Brazil. She complained of vertigo and presented skin lesions on her upper and lower limbs. She denied local trauma or insect bites and had no recent contact with cats. The patient was a current smoker (an average of ten cigarettes/day), used to drink an average of three cans of beer/day (discontinued for three months), and had insulin-dependent diabetes (diagnosed ten years earlier).

At admission, the patient was hemodynamically stable, had regular breathing, and was afebrile but presented thrombocytopenia and normochromic anemia (Table 1). The erythematous lesions had showed purulent exudate on both arms and the left leg for the last two months. According to the patient, the lesions initially appeared on the right upper limb, and 2 weeks later, they had spread to the left lower and upper limbs. The patient had been previously treated with amoxicillin-clavulanate, ciprofloxacin, cefepime, and topical silver sulfadiazine. On admission, the patient was treated with piperacillin/tazobactam for 12 days. Blood tests were negative for HIV, viral hepatitis, leishmaniasis, rheumatoid factor, and antinuclear antibodies. In addition, there was a polyclonal increase in gamma globulin (Table 2). A skin biopsy was performed on the sixth day of hospitalization, and a histological analysis showed a dense neutrophilic infiltrate and multinucleated giant cells. No fungi or protozoa were visualized, and the histological findings were compatible with pyoderma gangrenosum.

Further, the patient presented hepatosplenomegaly on the 14th day. After 20 days, she showed fatigue, weakness in the lower limbs, persistent fever (38 °C to 39 °C), odynophagia, inappetence, and no clinical improvement of the skin lesions. Blood tests revealed thrombocytopenia, leukopenia, and anemia. Following this, an empirical broad-spectrum antibiotic therapy (vancomycin, meropenem, and polymyxin B) was prescribed. A multidrug-resistant *Pseudomonas aeruginosa* was isolated from the blood, and only polymyxin B was maintained for the next 12 days.

Bone marrow exams were performed on the 23rd day and showed a hypercellular marrow with a slight predominance of the erythrocyte series, suggesting a reactive marrow. Histological sections showed a fragment of the iliac crest with normocellular functioning myeloid tissue, with usual cellular representatives of the three series. There was an absence of myelofibrosis, granulomas, parasitic agents, or primitive or metastatic neoplasm. A blood exam on the 28th day revealed low fibrinogen, slightly elevated ferritin, and pancytopenia (Table 1). In addition to hepatosplenomegaly, hemophagocytic syndrome was suggested.

Despite the above efforts, the patient presented hemodynamic instability, hypoxia (SpO2: 89%), and abdominal distention. The bone marrow fungal culture revealed *S. brasiliensis.* Sadly, this late result was obtained after the patient’s death (on the 32nd day of hospitalization), so no antifungal treatment could be prescribed.

The diagnosis of *S. brasiliensis* was made by a culture of the bone marrow aspirate in Sabouraud Dextrose Agar (SDA) for ten days at 25 °C (Figure 1A). Micromorphological hyphal characteristics in SDA—such as septum, pigmentation, and conidiogenesis—were observed (Figure 1B). The dimorphism was assessed by culturing the filamentous fungus at 37 °C, 5%CO_2_ for seven days in Brain Heart Infusion (BHI) agar medium. Cigar-like yeast cells were observed (Figure 1C,D). Subsequently, *S. brasiliensis* was identified by PCR (using the Sbra-F and Sbra-R primers) and electrophoresis, showing a 469 bp DNA fragment, as previously described by Rodrigues et al. 2015 [8]. Additionally, amplification and sequencing of the partial calmodulin gene were performed with degenerated primer pairs, as described by O’Donnell et al. (2000) (GenBank number OP811261) [9]. A broth microdilution test was performed to determine the minimum inhibitory concentrations (MIC) of itraconazole and amphotericin B (Sigma-Aldrich Co., USA), according to the CLSI protocol document M27 A3 for yeast (2008) [10]. RPMI 1640 medium was supplemented with 2% glucose; the final yeast concentration was increased to 1.0–5.0 × 10^5^ c.f.u. mL^−1^ and the yeasts were exposed to itraconazole or amphotericin B concentrations ranging from 0.03 to 16 μg mL^−1^ for 72 h at 37 °C. The MIC values corresponded to concentrations that inhibited ≥50% for itraconazole and 100% for amphotericin B of fungal growth, relative to the untreated control. The results are representative of two independent experiments made in duplicate. Quality control was performed with *Candida parapsilosis* ATCC 22019. The MIC values were itraconazole (1 µg/mL) and amphotericin B (2 µg/mL).

## 3. Discussion

The increasing number of *S. brasiliensis* infections in Brazil contributes to atypical and systemic sporotrichosis, which is frequently reported in immunosuppressed patients [1,4]. This case report reinforces the importance of systemic sporotrichosis in HIV-negative patients.

Usually, the clinical presentations of sporotrichosis are ulcerated cutaneous lesions that vary according to the host’s immune status [2,7]: (i) in acute cases, patients present increased inflammation with neutrophils and macrophage recruitment; and (ii) chronic lesions characterized by epithelioid or tuberculoid granulomas, lymphocytes, necrosis, and fibrosis [4,7]. In this context, the histological examination may not be enough to detect atypical sporotrichosis. In addition, stains, such as the periodic acid Schiff (PAS) or methenamine silver (Grocott), may have low sensitivity to detect fungal structures when compared to fungal culture [2,4,7].

Furthermore, the skin lesions can mimic other infectious diseases such as mycobacteriosis, actinomycosis, tegumentary leishmaniasis, blastomycosis, cryptococcosis, and paracoccidioidomycosis [4,11]. Although rare, there are a few reports of *Sporothrix* spp. infections mimicking pyoderma gangrenosum, presenting a large ulcer without lymphocutaneous spread [12,13,14]. In addition, for the management of pyoderma gangrenosum, ruling out the diagnosis of fungal infection before treatment with immunosuppressive drugs is crucial [12].

In systemic sporotrichosis, skin lesions may be associated with the fungal infection of other organs [6]. Besides pyoderma gangrenosum, the possibility of secondary hemophagocytic syndrome was also considered later in the course of the disease due to persistent fever, unresponsive to antibiotics, worsening cytopenia, and hepatosplenomegaly [15]. This condition is frequently associated with viral and bacterial infections; however, there are a few reports of this syndrome due to fungal infections (*Cryptococcus* spp., *Candida* spp., and *Histoplasma* spp.) [16]. Knowing the clinical manifestations and diagnostic criteria of hemophagocytic syndrome is important to initiate prompt therapy and improve the prognosis [15]. Interestingly, bone marrow alterations (aplasia) [6,17]; hematological alterations (anemia [4,6,17] and leukopenia [6,17]); gammopathies [4]; and hypoalbuminemia [4] are commonly seen in patients with systemic sporotrichosis. In this context, hematological and biochemical laboratory tests may help in diagnosing systemic manifestations of sporotrichosis [4], especially in the presence of widespread cutaneous lesions.

In the present case, the fungal identification was finished 17 days after collecting the bone marrow. The identification of *S. brasiliensis* was performed by molecular techniques. Fungal culture is considered the reference method for diagnosing sporotrichosis. However, in systemic manifestations that require immediate therapeutic intervention, fungal culture has proven to be an inefficient method, mainly due to the time needed for fungal growth [4,11]. It is crucial to establish faster and more accurate molecular methods for systemic sporotrichosis to be performed alongside fungal culture. PCR for *Sporothrix* species identification from clinical specimens would be useful by shortening the diagnostic time [18]. In addition, due to the broad spectrum of clinical manifestations of sporotrichosis, the use of serological methods in diagnostic screening favors rapid results with high levels of specificity and sensitivity [19]. Altogether, actions to favor the rapid diagnosis of systemic sporotrichosis are necessary. In addition, an earlier bone marrow culture could have been useful to achieve an early sporotrichosis diagnosis.

No clinical evidence of other types of immunosuppression was described for the patient. However, the patient had diabetes and reported daily alcohol drinking (discontinued for three months). Diabetes and alcoholism have been described to predispose to systemic sporotrichosis [2,12]. The effects of alcohol abuse and uncontrolled diabetes in affecting the cellular and humoral immunity is well established [20,21].

This case report demonstrates the challenge of diagnosing and managing severe systemic sporotrichosis with atypical manifestations in an HIV-negative patient, mainly considering the absence of epidemiological information. In this context, the transmission mode remains uncertain, since the patient denied contact with cats or organic material. In a region with epizootic feline sporotrichosis, like the one from this patient, *S. brasiliensis* is the most prevalent in human cases due to zoonotic transmission [4]. Otherwise, infection from the environment is uncommon, since recent works have pointed out that the soil works as a reservoir of fungal propagules due to latent fungi from the feces and carcasses of cats buried directly in the soil [3,4].

Therefore, we emphasize the need for more studies to identify severe sporotrichosis’ predisposing factors, other than HIV, to help with diagnosis and treatment. This study also highlights the need to develop novel strategies for differential diagnosis and the awareness of atypical cases of subcutaneous infections.

## Figures and Tables

**Figure 1 pathogens-11-01516-f001:**
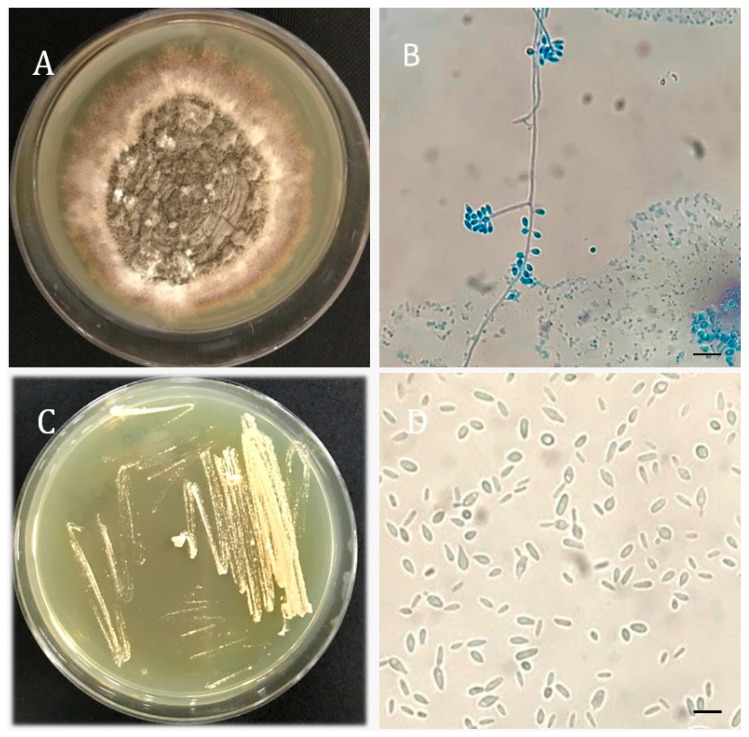
*Sporothrix brasiliensis* morphology. (**A**) Culture of filamentous colonies after ten days at 25 °C on Sabouraud Dextrose Agar (SDA). (**B**) Optical microscopy of hyphae and typical *S. brasiliensis* conidia (cotton blue, ×400). (**C**) Culture of yeast colonies after seven days at 37 °C and 5%CO_2_ on Brain Heart Infusion agar (BHI). (**D**) Optical microscopy of yeast colony (saline, ×400). Bar = 10 µm.

**Table 1 pathogens-11-01516-t001:** Laboratory parameters measured during the hospitalization of the disseminated sporotrichosis patient.

Laboratory Findings	Hospitalization Day									
	1	8	18	20	22	24	26	28	29	31
Leukocytes (/mm^3^)	4600	3770	**2310**	**1670**	**2730**	**2900**	**2650**	**2230**	**2020**	**1700**
Young neutrophils (%)	0	0	**6**	**17**	**8**	**4**	**6**	**8**	**26**	**33**
Erythrocytes (10⁶/mm^3^)	**3.38**	**3.16**	**2.86**	**2.64**	**2.54**	**2.5**	**2.71**	**2.83**	**2.66**	**2.71**
Hemoglobin (g/dL)	**10.5**	**9.7**	**8.7**	**7.8**	**7.7**	**7.5**	**8.3**	**8.6**	**8.3**	**8.2**
Thrombocyte (/mm^3^)	**65,000**	**73,000**	**49,000**	**36,000**	**43,000**	**45,000**	**52,000**	**31,000**	**40,000**	**28,000**
C-reactive protein (mg/L)	**54**	**37.2**	**49.1**	**40**	**47.1**	**31.5**	**33**	**39.5**	NP	**57.9**
Lactate (mmol/L)	NP	**3**	1.7	1.8	2.4	NP	NP	NP	NP	**3.5**
Glucose (mg/dL)	**188**	**197**	**124**	**147**	**240**	NP	NP	NP	NP	71
Urea (mg/dL)	19	30	39	25	24	NP	25	NP	NP	**60**
Creatinine (mg/dL)	0.48	0.31	0.34	0.37	0.43	NP	0.45	NP	NP	**0.77**
Aspartate aminotransferase (U/L)	**55**	NP	NP	NP	NP	NP	NP	NP	**114**	**118**
Alanine aminotransferase (U/L)	**36**	NP	NP	NP	NP	NP	NP	NP	**50**	**48**
Gamma-glutamyl transferase (U/L)	**417**	**417**	NP	NP	NP	NP	NP	NP	NP	NP
Alkaline phosphatase (U/L)	NP	**223**	NP	NP	NP	NP	NP	NP	NP	NP
Bilirubin (mg/dL)	0.75	NP	NP	NP	NP	NP	0.79	NP	1.04	**1.8**
Ferritin (ng/mL)	NP	**422.2**	NP	NP	NP	NP	NP	**294**	NP	NP
Iron (mcg/dL)	NP	**47**	NP	NP	NP	NP	NP	NP	NP	NP
Reticulocytes (%)	NP	**5.3**	NP	NP	NP	NP	NP	NP	NP	NP
Transferrin Saturation Index	NP	21%	NP	NP	NP	NP	NP	NP	NP	NP
Albumin g/dL	3.1	**2.8**	NP	NP	NP	NP	NP	NP	NP	NP
Fibrinogen (mg/dL)	NP	NP	NP	NP	NP	NP	NP	**125**	NP	NP
Blood culture	Negative	NP	NP	**Positive ***	NP	NP	NP	NP	NP	Negative

NP: not performed on that day. * Positive for Pseudomonas aeruginosa. In bold, parameters considered altered in the patient.

**Table 2 pathogens-11-01516-t002:** Serum protein electrophoresis measured on the 8th day of hospitalization of the patient.

Serum Protein Electrophoresis	
Total proteins g/dL	6.9
Albumin (%)	34.6
Alpha-1 (%)	5.9
Alpha-2(%)	9.7
Beta-1 (%)	5.5
Beta-2 (%)	7
Gamma (%)	**37.3**
Albumin/globulin ratio (%)	0.53
Monoclonal protein g%	Absent

In bold, parameters considered altered in the patient.

## Data Availability

Not applicable.

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
