# Peer review of "Late Diagnosis of Disseminated Sporothrix brasiliensis Infection with Bone Marrow Involvement in an HIV-Negative Patient"

_pathogens, 2022, doi:10.3390/pathogens11121516_

Round 1

Reviewer 1 Report

I am always pleased to review clinical case reports that are supported by basic mycology. This is a well-organized and interesting report by the authors. However, the manuscript still needs work; I have indicated my questions and suggestions for edits in the annotated PDF. In addition, I recommend that the authors have the manuscript reviewed for readability by an English language service or a native speaker of English. The language use is not grammatically incorrect, but currently sports some odd turns of phrases.

Author Response

We thank the reviewer for the suggestions. A thorough English review was performed in the manuscript. Below, is the point by point reply.

Line 32: needs reference for cats.

Response: Reference added.

Lines 32-34: please rewrite the sentence clarifying "other species"

Response: Sentence was rewritten to avoid misunderstanding. A phrase about the species was also included.

Line 38: Please change the highlighted to "CD4+ T"

Response: Thanks for your suggestion. Modified accordingly (Line 40).

Line 38, 41: Please use consistent terminology to avoid reader misunderstanding. e.g., "systemic" vs. "hematogenous" dissemination, which are referring to the same phenomenon.

Response: Thanks for your suggestion. Modified accordingly. We also included other factors associated to systemic sporotrichosis, as mentioned by the referee (Line 41-43).

Line 43: Because you established that the disease occurs in the low CD4+ T-cell milieu of HIV+ patients in the previous paragraph, please add an explanatory / descriptive line accommodating the possibility of disease in HIV- subjects BEFORE you launch into the case report.

Response: Thanks for your suggestion. Modified accordingly (Line 41-58).

Lines 51-54: Please add if the patient was a smoker and beer-drinker at the time of admission and if not, when they ceased to partake.

Response: Thanks for your suggestion. Information added accordingly (Lines 68-70).

Line 62: "polyclonal increase" when? increase compared to which baseline, tested when?

Response: Thanks for your suggestion. Information added accordingly. Table 2 was included showing the serum protein electrophoresis. Increase was related to refence values (9.2 – 19.8%).

Line 75: What was the reason for not performing bone marrow exams prior to the 23rd day? Would an earlier examination have helped the index of suspicion? I suggest the authors put in a line or two to comment on this aspect in the discussion.

Response: Since the multidrug-resistant Pseudomonas aeruginosa was isolated from blood, clinicians didn’t perform bone marrow exams before 23rd day. Yes, we think that an earlier bone marrow examination would be important for the sporothricosis diagnosis. A phrase about this was included in the discussion (Line 206-212, and Line 224-225).

Line 89: The use of "during" here is confusing. Please consider rephrasing.

Response: Thanks for your suggestion. Modified accordingly.

Line 92: observed from where? The initial SDA culture plate at day 10 or following further 7 days' growth in BHI medium?

Response: From 10-days SDA culture. This was included in the text (Line 105).

Line 100: please clarify what kind of modifications were made to the CLSI M27-A3 protocol.

Response: Thanks for your suggestion. The modifications to the protocol were included in the text (Line 122-132).

Line 124: You are describing culture as the gold standard; please put in some numbers for sensitivity of this method. In your own methodology, the culture-based method seems to have taken a total of 17 days. Is that optimal for decision making for patient treatment?

Further, your final diagnosis and identification was done using PCR-based techniques followed by gel electrophoresis and sequencing. Therefore, please consider qualifying your statement of culture being the gold standard for identification of S. brasiliensis.

Response: Thanks for your suggestion. We included a discussion about the suitability of culture methods for the diagnosis, and the need for standard and faster methods for systemic fungal infections (Line 213-224).

Line 129: If the published description of another case is so similar to the current report, what new information does the latter offer? I suggest the authors rephrase this paragraph. I would also suggest less frequent usage of conjunctive adverbs, such as "however" and "Otherwise" etc.

Response: Thanks for your suggestion. We modified the sentence, and we performed a thorough review of the conjunctive adverbs in the text.

Line 133: The authors correctly indicate that no clinical evidence of immunosuppression was described for the patient, with the proviso that such "clinical evidence" was restricted to certain blood cell counts and related parameters. By now it is well established that, for example, diabetes and alcoholism both cause immune suppression and impairment of host defences (see PMID: 19519161; PMID: 19630706). Therefore, I suggest revising this declarative statement.

Response: Thanks for your suggestion. We modified this statement in the text (Line 226-230).

Line 142: I would suggest that the authors incorporate in the text (in introduction or discussion) the fact that S. brasiliensis is not the only etiological agent for human sporotrichosis. The role of S. schenckii is well known.

Response: Thanks for your suggestion. This information was included in the text (Line 34-37).

Reviewer 2 Report

The manuscript “Late diagnosis of disseminated Sporothrix brasiliensis infection with bone marrow involvement in an HIV-negative patient” has high merit for reported a case of systemic sporotrichosis in an HIV-negative patient, with widespread cutaneous lesions mimicking pyoderma gangrenosum. It is an attractive, well-conducted, and structured manuscript containing essential information for health workers. Some minor considerations are described below: 

Page 3, line 8: Edit the citation by "Rodrigues et al 2015".

Page 3, line 13: Describe the modifications implemented in the broth microdilution test.

Page 4, line 18: Add the year of citation.

Page 4, line 32: I suggest, if possible, adding the incidence of sporotrichosis in the Minas Gerais state according to the last governmental estimations.

Author Response

We thank the reviewer for the suggestions. Below, there is the point by point reply.

Page 3, line 8: Edit the citation by "Rodrigues et al 2015".

Response: Thanks. The reference was edited.

Page 3, line 13: Describe the modifications implemented in the broth microdilution test.

Response: Thanks for your suggestion. We added the CLSI M27-A3 protocol and modifications (line 102-112).

Page 4, line 18: Add the year of citation.

Response: The paragraph was rewritten.

Page 4, line 32: I suggest, if possible, adding the incidence of sporotrichosis in the Minas Gerais state according to the last governmental estimations.

Response: We included a systematic review by Rabelo et al. (2022) which places Minas Gerais in the range of 101-1000 cases of sporotrichosis in the period of 1907–2020 (line 44-48).